# Object Concepts Emerge from Motion

**Haoqian Liang**[1], **Xiaohui Wang**[1], **Zhichao Li**, **Ya Yang**[1], **Naiyan Wang**

[1]Beijing University of Posts and Telecommunications

lianghq@bupt.edu.cn, nekomio@bupt.edu.cn, leeisabug@gmail.com,
yangya@bupt.edu.cn, winsty@gmail.com

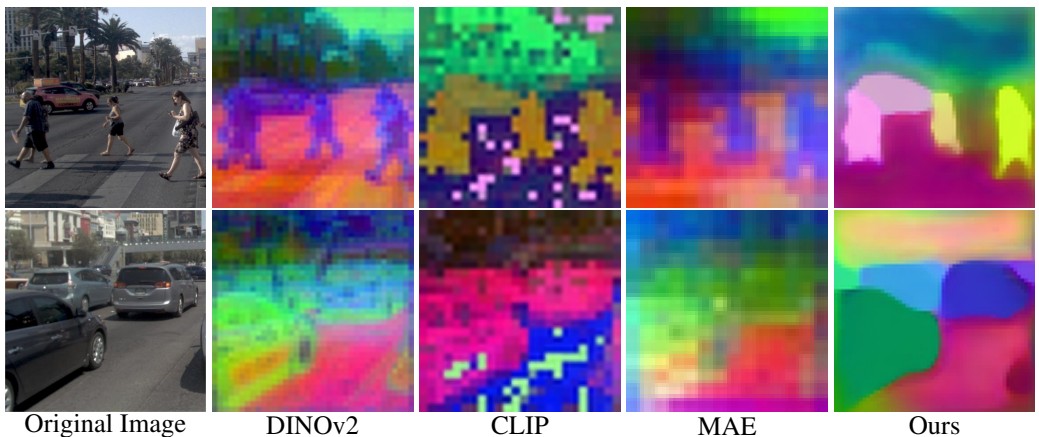

|  Original Image | DINOv2 | CLIP | MAE | Ours |

Figure 1: Comparisons with feature maps learned by our method and different visual foundation models. Our method focuses on the unity of object instance, in contrast to other methods emphasize on object class more.

## Abstract

Object concepts play a foundational role in human visual cognition, enabling perception, memory, and interaction in the physical world. Inspired by findings in developmental psychology—where infants are shown to acquire object understanding through observation of motion—we propose a biologically inspired framework for learning object-centric visual representations in an unsupervised manner. We were inspired by the insight that motion boundary serves as a strong signal for object-level grouping, which can be used to derive pseudo-instance supervision from raw videos. Concretely, we generate motion-based instance masks using off-the-shelf optical flow and clustering algorithms, and use them to train visual encoders via contrastive learning. Our framework is fully label-free and does not rely on camera calibration, making it scalable to large-scale unstructured video data. We evaluate our approach on three downstream tasks spanning both low-level (monocular depth estimation) and high-level (3D object detection and occupancy prediction) vision. Our models outperform previous supervised and self-supervised baselines and demonstrate strong generalization to unseen scenes. These results suggest that motion-induced object representations offer a compelling alternative to existing vision foundation models, capturing a crucial but overlooked level of abstraction: the visual instance. The implementation can be found here: https://github.com/yulemao/Object_Concepts_Emerge_from_Motion

# 1 Introduction

Physical AI aims to develop intelligent agents capable of perceiving and interacting with the physical world. A fundamental cognitive capacity required for such agents is the ability to recognize and understand the concept of "object"—a core unit of perception and reasoning. In the human visual system, the importance of object concepts is well-established in neuroscience. As noted by Kellman and Spelke [27], *"this cognitive ability not only supports object recognition and classification, but also plays a crucial role in spatial perception, memory formation, and the interaction between objects and their environment."* Understanding how object concepts are formed and represented in biological systems provides critical insights for building more robust and generalizable visual agents in artificial systems.

However, *what makes an object look like an object?* This is a non-trivial question, as objects can vary drastically in appearance, shape, and motion patterns. Early studies in developmental psychology [27] have demonstrated that the ability to perceive object unity is not innate, but learned during infancy. Infants begin to exhibit evidence of understanding object cohesion from around two months of age, with robust performance observed by four months. These findings suggest that object perception is a learned capacity grounded in sensory experience. Subsequent research [24, 40] has shown that motion cues—particularly common or coherent motion—serve as a powerful signal for infants to infer object boundaries and unity. As the visual system matures, this dynamic understanding is gradually internalized into the ventral visual stream [29, 53], enabling object recognition from static visual inputs alone. Inspired by this developmental trajectory, our work aims to design an unsupervised computational model that mimics this learning process: *beginning with motion-based interactions and evolving toward abstract, appearance-based object concepts.*

Recently, learning universal visual representations through self-supervised or weakly supervised paradigms has gained significant attention, due to their strong performance across a wide range of vision tasks. Among self-supervised approaches, notable examples include the DINO [7, 41] and MAE [23, 64] families, which rely on self-distillation and self-reconstruction mechanisms, respectively, to learn robust feature representations. Another influential direction leverages web-scale image-text pairs, as exemplified by CLIP [45], to align visual and language representations. To better understand what these models capture, we compare the low-dimensional PCA projections of features extracted by DINO, CLIP, and our model (see Fig. 1). We observe that DINO and CLIP tend to focus on semantic *categories*. However, neither method captures the concept of a semantic *instance*—a distinct, coherent object entity—adequately. We argue that existing visual foundation models overlook this crucial level of abstraction, which is fundamental for understanding the physical world.

In this work, we propose a biologically inspired framework for learning visual features that encode object-level semantics. As a first step, we explore this approach in outdoor driving scenarios, which provide rich motion cues arising from both ego-motion and independently moving objects. The key observation that inspires our method is that motion boundaries often align with object boundaries (detailed in Sec. 3.1), which echoes the discoveries in neuroscience that common motion is crucial to the early development of object unity. Based on this, we employ an off-the-shelf optical flow estimation algorithm, followed by a simple clustering technique, to generate pseudo instance masks without human supervision. These instance labels are then used to supervise representation learning via a contrastive objective. Importantly, unlike previous approaches [74, 4], our method does not require camera calibration parameters, allowing it to scale to large and diverse unlabeled video datasets.

Our motion-guided learning paradigm naturally bridges low-level and high-level vision. We validate our method on three downstream tasks: monocular depth estimation (low-level), 3D object detection and occupancy prediction (high-level). Across all model sizes, our method consistently outperforms supervised pretraining on ImageNet-22K and other self-supervised learning methods, demonstrating the effectiveness of learning object-centric representations from motion cues. Moreover, we find that our features are complementary to those from existing foundation models such as DINO—fusing them leads to further performance gains. Interestingly, although our model is trained only on outdoor scenes, it generalizes well to unseen indoor environments. This suggests that the learned features capture object composition and structure, rather than merely memorizing training-set appearances.

To summarize, our key contributions are as follows:

- We propose a biologically inspired paradigm for object-centric visual representation learning. Motivated by studies of infant cognition, we are the first to demonstrate the effectiveness and scalability of using motion as an unsupervised supervisory signal on a large-scale dataset and modern model architectures.
- We introduce a computationally efficient framework that implements this paradigm using off-the-shelf optical flow and simple clustering. Our approach scales naturally to large-scale outdoor driving datasets without requiring camera calibration or manual labels, and supports models of varying capacities.
- We extensively evaluate our models on three downstream tasks—monocular depth estimation (low-level), 3D object detection and occupancy prediction (high-level). Our method outperforms supervised and self-supervised methods across all model sizes, and shows strong generalization to unseen indoor scenes, highlighting the robustness and transferability of the learned object-centric features.

## 2 Related Work

### 2.1 Object Discovery

The central aim of object discovery is the identification and localization of objects within visual data, including images and videos, without the prerequisite of instance-level annotations for specific object classes. This paradigm significantly mitigates the need for large-scale, high-quality labeled datasets. Early approaches to object discovery included methods based on object occurrence frequency [25, 26, 54], and techniques utilizing region proposals to select key object bounding boxes through combinatorial optimization [52, 55, 56, 62, 75]. More recently, researchers have proposed numerous learning-based methods built upon the Transformer architecture. These approaches leverage features obtained from powerful pre-trained image models (e.g., DINO) to identify and segment objects via graph-based or spectral clustering techniques [46, 50, 59, 60, 73]. Another line of research adopts an object-centric perspective, frequently utilizing scene generation or reconstruction methodologies to derive learning signals, that involve decomposing scenes into their constituent parts (e.g., objects, background) and learning their respective representations [5, 18, 36, 38]. Similarly, another class of methods utilizes motion and multi-modal information as supervision, using the motion consistency of 2D or 3D points as a cue to distinguish objects from the background [34, 51, 61, 72]. To address the high demand for input dependencies of these works, our method only requires the simplest optical flow and clustering process to acquire object masks in an unsupervised manner. These masks subsequently serve as pseudo labels for single-image representation learning.

### 2.2 Visual Foundation Models

Visual foundation models aim to learn broadly applicable and transferable visual representations by pre-training on large-scale data. These learned general representations are intended to be transferred to downstream tasks by fine-tuning or prompting. Various self-supervised learning paradigms have been proposed for visual foundation models. Contrastive-learning-based methods pull together representations of different augmented views of the same image (positive samples) in an embedding space, while pushing apart representations of different images (negative samples) [8–10, 14, 22, 70]. Building upon this, subsequent self-distillation methods utilize "teacher" signals moving averaged by the student model itself for self-guided training, achieving excellent performance without relying on negative sample pairs [7, 12, 19, 41]. On the other hand, inspired by masked language modeling in natural language processing, masked-autoencoder-based methods learn representations by randomly masking portions of an image and training the model to reconstruct the masked content [2, 23, 44, 58, 64]. There is also some works that jointly learn the embeddings and predictions for more semantically rich and general features [1, 3]. Furthermore, despite different supervisory approaches, methods employing weak supervision, such as through text, have also made significant progress in the field of foundation models [45]. However, as illustrated in Fig. 1, all these pretrained models provide *semantic class* features rather than *semantic instance* features. We argue that semantic instance features could be beneficial for downstream tasks that require instance-level separation, such as object detection.

## 3 Method

To build an object-centric visual representation that generalizes across tasks and environments, we propose a biologically inspired learning framework centered on motion cues. Rooted in cognitive

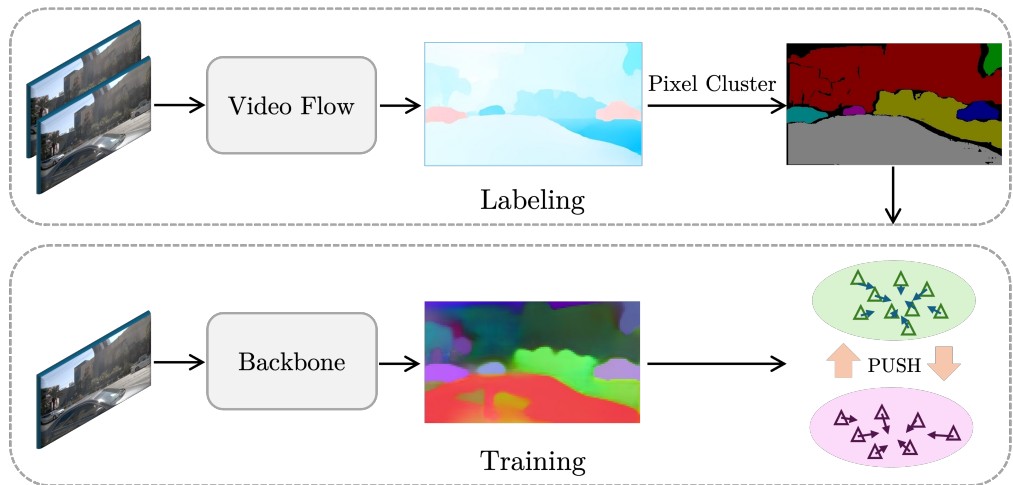

Figure 2: Pipeline of the proposed method.

developmental insights, our approach leverages the observation that coherent motion often indicates objecthood—an idea supported by infant perception studies and geometric reasoning in dynamic scenes. In this section, we introduce our method, which consists of three key components: (1) a geometric analysis revealing how motion boundaries correlate with object boundaries, (2) a data processing pipeline that extracts motion-induced pseudo-labels from large-scale video data, and (3) an unsupervised training objective designed to learn robust and transferable features from these labels. Together, these components form a scalable and calibration-free paradigm for learning object-level semantics from raw videos. The whole pipeline is shown in Fig. 2.

### 3.1 Geometric Insights

A central insight of our approach is that *motion boundaries often align with object boundaries*. It is obvious that if the object itself moves, its flow boundary can naturally serve as the object boundary. In this section, we provide a geometric and mathematical justification for why ego-motion can also be used to separate different objects under the assumption of rigid scenes.

Let $\mathbf{p} = (u, v)$ denote a pixel in the image domain, and $D(u, v)$ its corresponding depth. Assuming a pinhole camera model with intrinsic matrix $\mathbf{K}$, $\mathbf{p}$ is the pixel projected by the 3D point $\mathbf{P}$. Under rigid motion, the 3D scene point undergoes a transformation via camera pose change $(\mathbf{R}, \mathbf{t}) \in SE(3)$, resulting in a new image projection $\mathbf{p}'$ in the next frame. Projecting $\mathbf{P}'$ back into the image plane yields:

$$\begin{bmatrix} u' \\ v' \\ 1 \end{bmatrix} \sim \mathbf{K} \cdot \mathbf{P}' = \mathbf{K} \left( \mathbf{R} \cdot D(u, v) \cdot \mathbf{K}^{-1} \begin{bmatrix} u \\ v \\ 1 \end{bmatrix} + \mathbf{t} \right) \tag{1}$$

The optical flow is then computed as the pixel displacement. This means that the optical flow $\mathbf{F}(u, v)$ is a function of the depth $D(u, v)$, the camera motion $(\mathbf{R}, \mathbf{t})$, and the camera intrinsics $\mathbf{K}$. We summarize this dependency as:

$$\mathbf{F}(u, v) = \begin{bmatrix} u' - u \\ v' - v \end{bmatrix} = \phi(D(u, v); \mathbf{R}, \mathbf{t}, \mathbf{K}) \tag{2}$$

Taking the spatial gradient of the flow field gives:

$$\nabla \mathbf{F}(u, v) = \frac{d\phi}{dD} \cdot \nabla D(u, v) \tag{3}$$

This expression indicates that discontinuities in the flow field—i.e., motion boundaries—can arise from large gradients in the depth map. Under the assumption of rigid motion, these motion boundaries serve as reliable proxies for object boundaries. This geometric insight underpins our approach of utilizing motion cues to derive instance-level supervision.

This concept aligns with foundational theories in computer vision. David Marr, in his seminal book [39], articulated that:

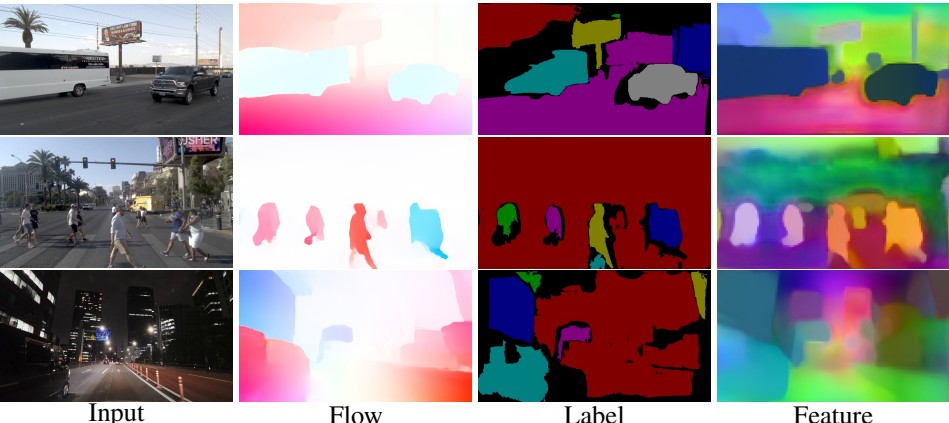

| Input | Flow | Label | Feature |

Figure 3: Examples of the pseudo-label generation results and the output features.

*"...the velocity field of motion in the image varies continuously almost everywhere, and if it is ever discontinuous at more than an isolated point, then a failure of rigidity (like an object boundary) is present in the outside world. In particular, if the direction of motion is ever discontinuous at more than one point—along a line, for example—then an object boundary is present."*

A notable advantage of our method is its independence from camera calibration. The necessary geometric information is inherently encoded within the optical flow, enabling us to train on large-scale, uncalibrated video datasets. This approach enhances scalability and broadens the applicability of our framework across diverse real-world scenarios.

## 3.2 Data Processing

**Data Sources.** We use two datasets in our approach: OpenDV-YouTube [67] and nuPlan [20]. Both datasets provide a large amount of high-quality and diverse unlabeled video data. OpenDV-YouTube contains videos collected from more than 244 cities all over the world, resulting in a total of 1747 hours of front-view videos. nuPlan provides 8 different camera views. It collects 1200 hours of driving data from 4 cities, 120 hours of which were recorded with 8 different camera views. We merged the two datasets and obtained approximately 2,700 hours of raw video data in total.

**Optical Flow Estimation.** We apply the pipeline of VideoFlow [49] to extract optical flow information from videos. The model takes five frames as input and outputs the optical flow for the middle three frames. We sample each clip with 0.3s intervals, and for each clip, we select the first frame within two consecutive frames as input, which spans 1 second.

**Pixel Cluster.** For all optical flow data generated by VideoFlow, we perform a simple Breadth-First Search (BFS) to cluster objects. Our algorithm takes the optical flow, the forward-backward consistency check result, and two thresholds $\theta_f$ and $\theta_s$ as input. For each pixel that satisfies the forward-backward consistency check, all neighboring pixels with a flow difference smaller than $\theta_f$ are considered to belong to the same object. $\theta_s$ is the minimum number of pixels to form a cluster. Pseudo-code of our algorithm is provided in the appendix.

**Results.** We set two thresholds to $\theta_f = 1.5, \theta_s = 100$. Fig. 3 shows some example results. As illustrated in the pseudo-label visualizations, the proposed algorithm successfully segments objects exhibiting significant movement, such as moving cars and pedestrians. Furthermore, because objects at different depths exhibit different apparent motion in the image even when they are stationary, the proposed algorithm is also able to segment foreground instances such as trees and signs. The pseudo-labels exhibit under segmentation due to weak motion cues or errors in optical flow estimation. Such cases are explicitly handled in the design of the loss function. We retained all samples with at least two pseudo-labels(i.e., at least one foreground cluster) and successfully obtained a total of 48 million images along with their corresponding pseudo-labels for pre-training.

## 3.3 Pre-training

**Overall Structure.** Our network architecture follows the design proposed in [28]. The network takes one image as input. Due to the need for high-resolution feature maps, we choose backbone

networks(*e.g.*, ResNet [21], Swin [35]) whose computational cost scales linearly with input size. These features are then processed by a Feature Pyramid Network (FPN). Similar to the semantic segmentation branch in [28], the information from all levels of the FPN pyramid is merged into a simple output. The resulting feature map has a spatial resolution of 1/4 the input size and a channel dimension of 64. A 2× bilinear upsampling is then applied, and each feature vector is normalized to yield the final output features.

**Training Loss.** Based on the labels derived from the optical flow and the output feature map, we design a simple yet effective loss function. As discussed in Sec. 3.2, the pseudo-labels can only extract a subset of the instances, making it inappropriate to cluster all background regions together. Since the number of background pixels is usually significantly larger than that of instance pixels, we treat the label with the highest pixel count as the background. The loss between any two background pixels is ignored. The loss function between two pixels $i$ and $j$ is defined as follows:

$$L(i,j) = \begin{cases} \|F_i - F_j\|_2^2, & y_i = y_j \neq 0 \\ \max\{m - \|F_i - F_j\|_2, 0\}^2, & y_i \neq y_j \\ 0, & y_i = y_j = 0 \end{cases} \quad (4)$$

where $F_i$ and $F_j$ are the feature vectors of the final output feature map of the network, $m$ is a margin parameter, $y$ represents the instance label derived from the optical flow. $y = 0$ denotes the background. We set the margin parameter $m$ to 1.0 in our implementation.

The total loss over all sampled pixel pairs is defined as:

$$L_{total} = \frac{1}{N} \sum_{i,j} L(i,j). \quad (5)$$

## 4 Experiments

To validate the effectiveness of our method across the vision spectrum, we conduct comprehensive experiments on both low-level and high-level vision tasks. Our core hypothesis is that the model, by learning from low-level cues, develops an internal understanding of object composition, which subsequently benefits high-level semantic reasoning. Conversely, this object-centric representation also enhances performance on low-level tasks by providing richer contextual cues. We evaluate our models on three representative downstream tasks: monocular depth estimation (low-level), 3D object detection, and 3D occupancy prediction (high-level).

### 4.1 Implementation Details

We implement the proposed method using PyTorch [43] and mmPretrain [11]. We train models on Swin Transformer [35] (Tiny to Large) and ResNet-50 [21]. All Swin models use a window size of 7, while the B and L variants of SimMIM [64] and Semantic-SAM [30], which are used for comparison, adopt a larger window size of 12. This larger window is usually beneficial due to the increased context, at the expense of higher computational cost. AdamW optimizer [37] with a weight decay of 0.05 is adopted. All models are trained for 200 epochs using a cosine decay learning rate scheduler and 10 epochs of linear warm-up. The initial learning rate is set to 0.001 and batch size is set to 2048. All input images are cropped and resized to a resolution of $224 \times 224$. We employ a data augmentation strategy that includes random flipping, brightness, and gamma adjustment. We sample 200 labeled pixels from each image for training. We further fine-tune the models for 20 epochs with an initial learning rate of $2 \times 10^{-5}$ and a weight decay of $10^{-4}$. During fine-tuning, two random crops are extracted from each input image, and the loss is calculated both within each crop and between the two. This fine-tuning process further enhances the separation of distant objects in large images. All downstream models are trained with official open-sourced code for comparison. During fine-tuning on downstream tasks, only the pretrained weights of the backbone are utilized for a fair comparison.

### 4.2 Qualitative Results

The fourth column in Fig. 3 shows PCA projections of our model's features. Thanks to the generalization of the backbone network, the features reveal a key strength: the model distinguishes many objects not annotated in the pseudo-labels—such as distant cars, pedestrians, and even static structures like buildings and poles. This suggests that our model goes beyond mimicking pseudo-labels and learns a more general, object-centric representation. Fig. 4 further visualizes similarity maps from selected

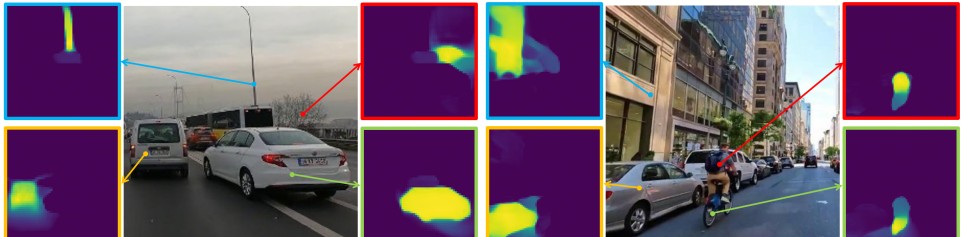

Figure 4: Similarity visualization for a set of reference points.

Table 1: Quantitative evaluation of DCDepth [57] on the KITTI Eigen split using different pretraining.

| Method | Backbone | SILog ↓ | Abs Rel ↓ | Sq Rel ↓ | RMSE ↓ | RMSE log ↓ | $\delta_1$ ↑ | $\delta_2$ ↑ | $\delta_3$ ↑ |
|---|---|---|---|---|---|---|---|---|---|
| ImageNet-22K | Swin-T | 7.455 | 0.055 | 0.165 | 2.182 | 0.082 | 0.969 | 0.996 | 0.999 |
| Semantic-SAM [30] | Swin-T | 7.346 | 0.055 | 0.165 | 2.169 | 0.082 | 0.971 | 0.996 | 0.999 |
| DINOv2 [41] | ViT-S | 7.119 | 0.052 | 0.158 | 2.153 | 0.079 | 0.974 | 0.997 | 0.999 |
| ImageNet-22K | Swin-L | 6.891 | 0.051 | 0.145 | 2.044 | 0.076 | 0.977 | 0.997 | 0.999 |
| Semantic-SAM [30] | Swin-L | 6.713 | 0.049 | 0.137 | 2.007 | 0.074 | 0.979 | **0.998** | **1.000** |
| SimMIM [64] | Swin-L | **6.542** | 0.048 | 0.130 | 1.941 | 0.073 | 0.979 | **0.998** | 0.999 |
| Ours | Swin-T | 6.991 | 0.051 | 0.145 | 2.016 | 0.077 | 0.975 | 0.997 | 0.999 |
| Ours | Swin-S | 6.736 | 0.049 | 0.138 | 1.981 | 0.075 | 0.978 | 0.997 | 0.999 |
| Ours | Swin-B | 6.598 | 0.048 | 0.131 | 1.939 | 0.073 | **0.981** | 0.997 | 0.999 |
| Ours | Swin-L | 6.558 | **0.047** | **0.129** | **1.929** | **0.072** | **0.981** | 0.997 | 0.999 |

reference points. The sharp boundaries and clear object separation confirm that our features capture consistent, instance-level semantics, even without explicit supervision.

### 4.3 Monocular Depth Estimation

We evaluate our model on the KITTI dataset [16] using the standard Eigen split [15], with DCDepth [57] as the decoder. As shown in Tab. 1, our model consistently outperforms both supervised ImageNet-22K pretraining and models pretrained on the Semantic-SAM [30], which is a weakly supervised method utilizing large-scale pseudo segmentation annotations.

Our approach achieves superior performance across all backbone sizes. For instance, with Swin-Tiny, our model reduces the RMSE to 2.016 (compared to 2.169 from Semantic-SAM) and improves the $\delta_1$ accuracy to 0.975. These results are even comparable to Swin-Large with ImageNet-22K pretraining. As the backbone scale increases, the performance of our method improves steadily.

As shown in Tab. 5, combining our features with DINO leads to consistent and significant performance improvements. While our method alone already outperforms using either DINO or ImageNet-22K pretrained features in isolation (rows 1–3), the best result is achieved when concatenating our features with DINO pretrained features (row 6), reaching the lowest SILog (6.796) and a competitive RMSE (2.014).

This highlights the complementary nature of the two representations: DINO focuses more on semantic category-level cues, while our method emphasizes instance-level object structure derived from motion cues. Fusing them allows the model to leverage both semantic context and object-centric information, leading to improved depth estimation performance.

Tab. 2 further shows the results on the official KITTI online leaderboard. Our method outperforms other methods in the primary metric (SILog) and also achieves competitive performance across the other evaluation metrics.

### 4.4 3D Object Detection

We evaluate our learned visual representations on the nuScenes dataset [6] for the 3D object detection task, using BEVFormer V2 [31, 66] as the detection framework. We compare our method against a diverse set of pretraining strategies, including supervised ImageNet-22K and COCO, as well as self-supervised approaches such as MoCo [10] and SimMIM [64].

As shown in Tab. 3, our approach achieves consistent and substantial improvements in both mean Average Precision (mAP) and NuScenes Detection Score (NDS) across multiple backbones. For instance, with a Swin-Tiny backbone, our model achieves an mAP of 43.01% and NDS of 52.41%,

Table 2: Quantitative results on the official split of KITTI dataset. All metrics reported here are from the KITTI online leaderboard.

| Method | Backbone | Pretrain | SILog ↓ | Abs Rel ↓ | Sq Rel ↓ | iRMSE ↓ |
|---|---|---|---|---|---|---|
| NeW CRFs [71] | Swin-Large | ImageNet-22K | 10.39 | 8.37 | 1.83 | 11.03 |
| VA-DepthNet [32] | Swin-Large | ImageNet-22K | 9.63 | 7.96 | 1.66 | 10.44 |
| IEBins [48] | Swin-v2-Large | MIM [65] | 9.84 | 7.82 | 1.60 | 10.68 |
| NDDepth [47] | Swin-v2-Large | MIM [65] | 9.62 | **7.75** | 1.59 | 10.62 |
| DCDepth [57] | Swin-Large | Semantic-SAM [30] | 9.60 | 7.83 | **1.54** | **10.12** |
| DCDepth [57] | Swin-Large | Ours | **9.54** | 7.76 | 1.55 | 10.37 |

Table 3: Quantitative evaluation of BEVFormerV2 [66] on nuScenes `val` set using different pretraining methods.

| Method | Backbone | NDS ↑ | mAP ↑ | mATE ↓ | mASE ↓ | mAOE ↓ | mAVE ↓ | mAAE ↓ |
|---|---|---|---|---|---|---|---|---|
| COCO | Res50 | 51.82 | 41.99 | 66.89 | 28.14 | 39.15 | 38.34 | 19.28 |
| ImageNet-1K | Res50 | 51.99 | 42.51 | **65.90** | 27.79 | 42.12 | **37.70** | **19.20** |
| MoCo v3 [10] | Res50 | 52.42 | 42.94 | 67.13 | 27.70 | **35.84** | 39.91 | 19.95 |
| Ours | Res50 | **52.55** | **43.22** | 66.30 | **27.56** | 37.76 | 38.47 | 20.53 |
| ImageNet-22K | Swin-T | 51.69 | 42.12 | 67.69 | **28.07** | **38.39** | 40.89 | 18.68 |
| Ours | Swin-T | **52.41** | **43.01** | **65.81** | 28.30 | 41.43 | **37.23** | **18.15** |
| ImageNet-22K | Swin-S | 53.62 | 45.22 | **64.94** | 27.75 | **36.97** | 40.07 | 20.18 |
| Ours | Swin-S | **54.22** | **45.49** | 65.22 | **27.73** | 37.61 | **35.61** | **19.04** |
| ImageNet-22K | Swin-B | 53.98 | 45.48 | 65.94 | 28.10 | 35.82 | 38.75 | **19.01** |
| SimMIM [64] | Swin-B | 54.03 | 45.18 | 63.81 | **27.53** | 38.02 | 37.04 | 19.22 |
| Ours | Swin-B | **55.68** | **47.54** | **62.74** | 27.84 | **33.79** | **36.77** | 19.81 |
| ImageNet-22K | Swin-L | 54.59 | 45.91 | 65.39 | 27.44 | 34.31 | 37.64 | 18.87 |
| SimMIM [64] | Swin-L | 54.98 | 46.52 | 64.80 | 28.06 | 33.72 | **35.87** | 20.36 |
| Ours | Swin-L | **55.80** | **47.29** | **62.83** | **27.16** | **33.20** | 36.50 | **18.77** |

outperforming the ImageNet-22K pretrained counterpart (mAP 42.12%, NDS 51.69%). As the backbone scales up to Swin-Large, our model further improves to 47.29% mAP and 55.80% NDS, still outperforming the compared supervised and self-supervised methods.

To compare with more methods based on ViT architectures whose computational cost are not affordable for high input resolution, we also tested various methods at a resolution of $704 \times 256$. As shown in Tab. 4, our Swin-based models achieve competitive or superior performance compared to DINOv2, while using significantly fewer parameters and lower computational costs. For instance, our model pretrained with the Swin-L backbone attains an NDS of 52.03% and an mAP of 41.79%. These results are comparable to those achieved by DINOv2 with the ViT-L backbone.

Notably, these improvements are not limited to Transformer-based architectures. With ResNet backbones such as R50, our model also outperforms COCO-supervised models, indicating that the benefit of our pretraining is architecture-agnostic. This broad compatibility with both convolutional and Transformer backbones highlights the generality of the learned features.

These gains can be attributed to the object-centric and geometry-aware priors introduced by our object-based visual representation. Unlike traditional supervised pretraining, our approach enables the model to internalize compositional structure and spatial relationships between objects. This proves particularly valuable in 3D detection tasks, where reasoning about object placement, extent, and occlusion is critical.

### 4.5 3D Occupancy Perception

We evaluate our method on the nuScenes validation set using SparseOcc [33] as the occupancy prediction framework. As shown in Tab. 6, our pre-trained models outperform both supervised (ImageNet-22K) and self-supervised (SimMIM) counterparts across all Swin backbone variants. Crucially, the strong performance of our method can be attributed to the underlying geometric insight described in Sec.3.1. By leveraging this property, our method is able to encode spatial structures that are semantically meaningful, even without direct instance-level annotations.

Table 4: Quantitative evaluation of BEVFormerV2 [66] on nuScenes `val` set using different pretraining methods at a resolution of $704 \times 256$

| Method | Backbone | NDS↑ | mAP↑ | mATE↓ | mASE↓ | mAOE↓ | mAVE↓ | mAAE↓ |
|---|---|---|---|---|---|---|---|---|
| DINOv2 | ViT-S | 46.24 | 34.88 | 71.65 | 28.45 | 49.97 | _42.70_ | _18.84_ |
| DINOv2 | ViT-B | _49.08_ | _38.36_ | _69.74_ | _28.21_ | _41.81_ | **42.40** | 18.84 |
| DINOv2 | ViT-L | **51.91** | **42.05** | 65.04 | 27.35 | **36.45** | 43.79 | **18.51** |
| ImageNet-22K | Swin-T | _47.42_ | _36.34_ | _70.90_ | _28.40_ | _48.36_ | **40.47** | **19.40** |
| Ours | Swin-T | **48.24** | **37.08** | **70.61** | **27.99** | **44.45** | _40.54_ | _19.45_ |
| ImageNet-22K | Swin-S | _48.78_ | _38.00_ | _70.65_ | _28.35_ | _41.20_ | 42.95 | _19.01_ |
| Ours | Swin-S | **50.87** | **40.23** | **68.88** | **27.85** | **40.45** | **36.67** | **18.61** |
| ImageNet-22K | Swin-B | _50.42_ | _40.71_ | 68.56 | **27.80** | _40.60_ | 42.81 | _19.52_ |
| Ours | Swin-B | **51.69** | **41.36** | **66.01** | 28.03 | **37.98** | 39.73 | **18.10** |
| ImageNet-22K | Swin-L | _50.48_ | _40.09_ | 68.01 | **27.90** | _40.69_ | _40.67_ | _18.38_ |
| Ours | Swin-L | **52.03** | **41.79** | 66.10 | 28.12 | **36.84** | 40.13 | **17.51** |

Table 5: Ablation studies on the KITTI depth estimation task. We evaluate the impact of different pretraining strategies: DINO refers to DINOv2 [41], and IN denotes ImageNet-22K [13] supervised pretraining. Ours and IN use Swin-T as the backbone, while DINO uses ViT-S.

| Ours | DINO | IN | SILog↓ | AbsRel↓ | RMSE↓ |
|---|---|---|---|---|---|
| ✓ | | | **6.991** | **0.051** | **2.016** |
| | ✓ | | _7.119_ | _0.052_ | _2.153_ |
| | | ✓ | 7.455 | 0.055 | 2.182 |
| | ✓ | ✓ | 7.071 | _0.052_ | 2.117 |
| ✓ | | ✓ | _6.927_ | **0.050** | **2.009** |
| ✓ | ✓ | | **6.796** | **0.050** | _2.014_ |

Table 6: Quantitative evaluation of SparseOcc [33] on nuScenes `val` set using different pretraining.

| Method | BB | RayIoU | RayIoU$_{1m, 2m, 4m}$ | | |
|---|---|---|---|---|---|
| MoCo v3 [10] | R50 | 34.4 | 28.3 | 35.1 | 39.9 |
| ImageNet-1K | R50 | _35.0_ | _28.8_ | _35.6_ | _40.5_ |
| Ours | R50 | **36.4** | **30.2** | **37.1** | **41.8** |
| ImageNet-22K | Sw-T | _35.5_ | _29.4_ | _36.3_ | _40.9_ |
| Ours | Sw-T | **37.0** | **31.1** | **37.8** | **42.2** |
| DINOv2 [41] | ViT-S | 35.9 | 29.5 | 36.8 | 41.4 |
| ImageNet-22K | Sw-S | _36.8_ | _30.4_ | _37.6_ | _42.3_ |
| Ours | Sw-S | **38.1** | **32.0** | **39.1** | **43.4** |
| DINOv2 [41] | ViT-B | 37.1 | 31.0 | 37.9 | 42.4 |
| ImageNet-22K | Sw-B | 37.6 | 31.3 | 38.4 | 43.1 |
| SimMIM [64] | Sw-B | _38.0_ | _31.7_ | _38.7_ | _43.4_ |
| Ours | Sw-B | **38.3** | **32.1** | **39.1** | **43.7** |
| DINOv2 [41] | ViT-L | **39.0** | **32.8** | **39.9** | **44.3** |
| ImageNet-22K | Sw-L | 37.6 | 31.4 | 38.4 | 43.0 |
| SimMIM [64] | Sw-L | 38.6 | _32.6_ | 39.4 | 43.7 |
| Ours | Sw-L | _38.7_ | _32.6_ | _39.5_ | _43.8_ |

## 5 Discussion and Future Work

### 5.1 Generalization to out of domain scenes

As a preliminary investigation, we train our method on outdoor driving videos. To demonstrate the generalization of our learned features, we also visualize the features on daily life videos from the Ego4D [17] and the robot manipulation dataset RTX [42], as shown in Fig. 5. Though not perfect, our model is capable of distinguishing different objects that did not appear in the training set. The first row shows some common scenes in indoor scenes. Our method can segment the unseen objects, including windows, tools, even cats, and hands. We observe similar results in the second row for robot manipulation. These results illustrate that our method does not overfit the objects in the training set, but indeed learns the essential composition of an object.

### 5.2 Further scaling up and extensions

We also attempted to apply our method to broader scenarios, such as ego-centric videos and unconstrained videos from the web. However, we found that the performance of our method is greatly limited by the performance of optical flow. Fortunately, we find that recent closely related work in monocular depth estimation [68, 69] improves significantly with the help of large-scale synthetic data. We hope a similar paradigm can also benefit the performance of optical flow.

Our method can be further extended to a temporal setting. Based on the compact object instance representation, we can easily endow the model with temporal prediction ability. It is essentially a world model that could predict the dynamics of the world. We will pursue these directions in our future work.

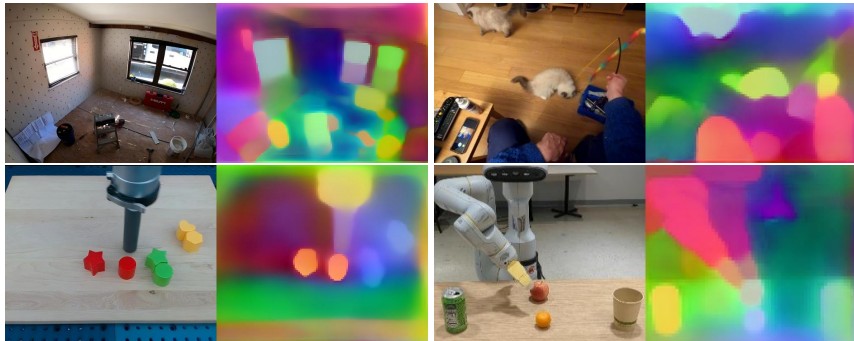

Figure 5: Examples of feature maps in out of domain scenes.

### 5.3 Improving precise localization ability

As seen in the visualization, our method focuses on the whole object, which means the features of the parts within an object are indistinguishable. This makes our method unsuitable for applications that need precise localization, such as keypoint matching. Our method can be improved by combining previous works that emphasizes local feature learning, such as CroCo [63], to get the best of two worlds.

## 6  Conclusions

In this work, we present a biologically inspired framework for learning object-centric visual representations, drawing motivation from developmental psychology studies on how infants acquire the concept of objects through motion cues. By leveraging the natural correlation between motion boundaries and object boundaries, our method derives instance-level pseudo labels from raw videos, enabling unsupervised representation learning without human annotations or camera calibration.

Through extensive experiments across three diverse vision tasks, we demonstrate that our approach not only matches but surpasses the supervised and self-supervised pretraining baselines. Our learned features capture object-level semantics that are complementary to those in existing vision foundation models such as DINO and MAE.

These results highlight the potential of integrating biologically inspired mechanisms—such as motion-guided grouping—into the design of scalable, general-purpose visual pretraining frameworks. We hope this work encourages further exploration of cognitive principles in building more robust and human-aligned vision systems.

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

# A Pseudo-codes for Pixel Cluster

For all optical flow data generated by VideoFlow, we perform a simple Breadth-First Search(BFS) to segment moving objects. Alg. 1 provides a pseudocode description of our algorithm. The algorithm takes the optical flow, the forward-backward consistency check result, and two thresholds $\theta_f$ and $\theta_s$ as input. $\theta_f$ is used to determine when the optical flow of two adjacent pixels, being sufficiently close, is considered to belong to the same object. $\theta_s$ controls the minimum number of pixels that an object should have.

---

**Algorithm 1** Pixel Cluster

---

**Input:** flow(optical flow), valid(consistency check), $\theta_f, \theta_s$
 1: Initialization:$n \leftarrow 0, v[i][j] \leftarrow false, S \leftarrow \emptyset$
 2: **for** $x \leftarrow 1$ to $H$ **do**
 3:    **for** $y \leftarrow 1$ to $W$ **do**
 4:       **if** $v[x][y] = true$ or valid$[x][y] = false$ **then**
 5:          continue
 6:       **end if**
 7:       $Q \leftarrow$ empty queue, $C \leftarrow \emptyset$
 8:       Enqueue$(Q, (x, y))$
 9:       **while** $Q \neq \emptyset$ **do**
10:          $(x, y) \leftarrow$ Dequeue$(Q)$
11:          $C \leftarrow C \cup \{(x, y)\}$
12:          **for** (i, j) in (x, y)'s 4 neighbors **do**
13:             **if** $||$flow$[i][j],$flow$[x][y]||_2 \leq \theta_f$ and $v[i][j] = false$ and valid$[x][y] = true$ **then**
14:                $v[i][j] = true$
15:                Enqueue$(Q, (i, j))$
16:             **end if**
17:          **end for**
18:       **end while**
19:       **if** $|C| \geq \theta_s$ **then**
20:          $S \leftarrow S \cup \{C\}$
21:       **end if**
22:    **end for**
23: **end for**
**Output:** $S$

---

# B Data Augmentation Details

All input images are first randomly resized to a resolution between $512 \times 288$ and $1024 \times 576$. They are then randomly cropped to $224 \times 224$. During cropping, up to 10 attempts are made to ensure that the cropped region contains at least two distinct labels. Afterward, each image has a 50% chance of being horizontally flipped. Additionally, gamma, brightness, and color augmentations are applied with a 50% probability, each sampled within the range of (0.9, 1.1).

# C More Qualitative Results

Fig. 6 shows additional qualitative results of the pseudo-label generation and the visualizations of the output features. As illustrated in the pseudo-label visualizations, the proposed algorithm successfully segments objects exhibiting significant movement, as well as foreground instances exhibiting motion patterns distinct from the background. The feature visualizations shows that the model distinguishes many objects not annotated in the pseudo-labels. This suggests our model goes beyond mimicking pseudo-labels, but learning a more general, object-centric representation.

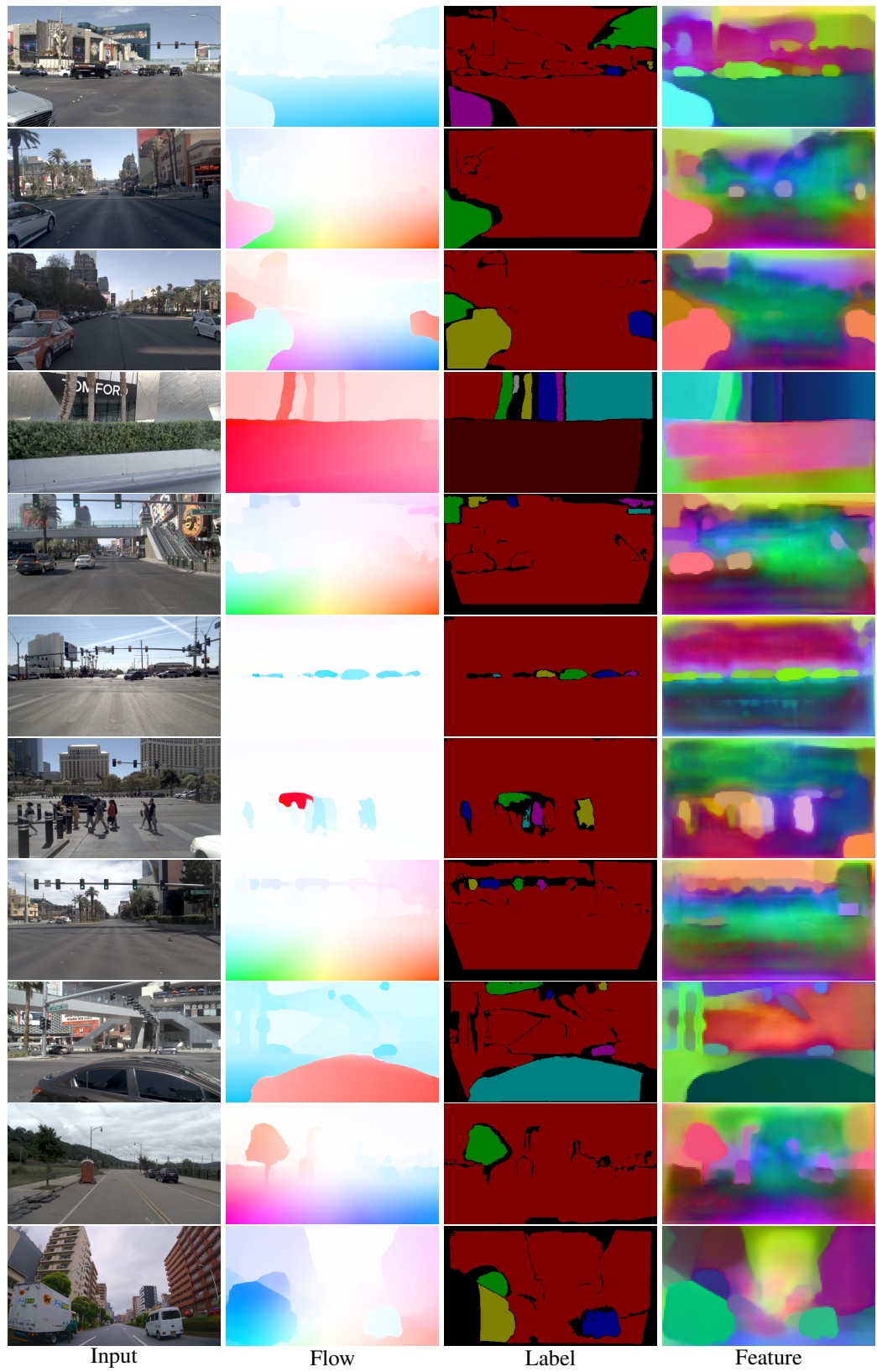

Input        Flow        Label        Feature

Figure 6: Examples of the pseudo-label generation results and the output features.

