# OpenReview forum: "Object Concepts Emerge from Motion"
_NeurIPS.cc/2025/Conference — NeurIPS 2025 poster_

### Official Review · Reviewer_Rb8Z · 2025-06-26

**Clarity:** 3
**Significance:** 3
**Originality:** 3
**Rating:** 5
**Confidence:** 4

**Summary:**

Inspired by insights from classic computer vision and developmental psychology regarding the importance of motion boundaries for inferring object boundaries and grouping image elements with similar motion into objects, the authors propose a system for unsupervised learning of "object-centric" visual representations. They leverage an off-the-shelf optic flow algorithm to generate motion-defined object instance masks and use them for contrastive visual representation learning. The approach is evaluated extensively and shown to outperform previous supervised and self-supervised baselines on three downstream tasks (monocular depth estimation, 3D object detection, occupancy prediction).

**Questions:**

For future work I'd invite the authors to think about replacing the off-the-shelf optic flow algorithm with a learning-based framework.

**Ethical Concerns:**

["NO or VERY MINOR ethics concerns only"]

**Final Justification:**

My impression of the paper is even stronger after the authors have thoroughly addressed various concerns in their rebuttal.

**Limitations:**

I did not see a particular mentioning of potential negative societal impact of this work. But I also don't see this as necessary.

**Paper Formatting Concerns:**

Looks good.

**Quality:**

3

**Strengths And Weaknesses:**

Strengths:
The approach is novel and outperforms several baselines on a set of diverse tasks comprising low-level (monocular depth estimation) and high-level (3D object detection, occupancy prediction) ones. I find it noteworthy that the approach even outperforms some supervised baselines. I feel the work is likely to lead to many follow-up works. As a work demonstrating strong representations emerging from unsupervised learning from large video datasets it is very timely and interesting.

I enjoyed the generalization to indoor scenes outside of the training domain.

Weaknesses:

1) I think some claims need to be toned down:
1a) Abstract:
"Our key insight is that motion boundary serves as a strong signal for object-level grouping". This is not at all a new insight. You are just using this very old insight in a novel way.

1b) Introduction:
"we are the first to leverage motion as an unsupervised supervisory signal to guide the emergence of object-level semantics in visual
representations." This statement also seems rather bold. There is a long history of works (pre-dating the deep learning era) of using motion to segment objects or infer the 3-D structure of the world. I think you should replace "emergence of object-level semantics in visual
representations" with something more specific.

2) You frequently use "Developmental Neuroscience" when "Developmental Psycholog" is more correct. (Developmental Neuroscience is typically studying the developing brains of animals with invasive techniques that cannot be applied to human infants and children.)

3) Occasional grammar problems
line 124: missing "that" or similar
line 141: used to separate or used for separating
line 181: exhibit
line 195: that -> whose. Or use: for which computational costs scale...

---

> ### Author Rebuttal · Authors · 2025-07-31
>
> Dear Reviewer Rb8Z,
> We greatly appreciate your valuable comments and questions. Our responses are as follows.
>
> ***Abstract: "Our key insight is that motion boundary serves as a strong signal for object-level grouping". This is not at all a new insight. You are just using this very old insight in a novel way.***
> Our idea builds upon numerous prior works on object discovery, and we did not intend to claim this as a novel insight. We will revise the sentence to: “We were inspired by the insight that motion boundary serves as a strong signal for object-level grouping, which can be used to derive pseudo instance supervision from raw videos.”
>
> ***Introduction: "we are the first to leverage motion as an unsupervised supervisory signal to guide the emergence of object-level semantics in visual representations." This statement also seems rather bold. There is a long history of works (pre-dating the deep learning era) of using motion to segment objects or infer the 3-D structure of the world. I think you should replace "emergence of object-level semantics in visual representations" with something more specific.***
> We agree that the current statement in the introduction may appear too strong. We fully acknowledge the rich body of prior work leveraging motion cues for object segmentation and 3D structure inference. We will revise it to “we are first to demonstrate the effectiveness and scalability of using motion as an unsupervised supervisory signal on a large-scale dataset and modern model architectures”
>
> ***You frequently use "Developmental Neuroscience" when "Developmental Psychology" is more correct.***
> Since our work mainly addresses cognitive aspects, we agree that 'Developmental Psychology' would be a more appropriate term. We will revise the text accordingly.
>
> ***Occasional grammar problems***
> We are grateful to the reviewer for pointing out these grammar issues. We will thoroughly review the manuscript and make the necessary corrections.
>
> ***For future work I'd invite the authors to think about replacing the off-the-shelf optic flow algorithm with a learning-based framework.***
> We sincerely appreciate your valuable suggestions regarding future work directions. In our current method, optical flow estimation is separated from model training. This was a deliberate choice to first test the basic effectiveness of our framework using a simple and reliable setup.
> However, we agree that an end-to-end learning framework is a promising direction. We are exploring ways to train the optical flow estimator together with our model, so that the main task can help improve flow estimation, and better flow can in turn improve the model’s performance and generalization.

---

> > ### Comment · Reviewer_Rb8Z · 2025-08-03
> > **Thank you for addressing my concerns**
> >
> > The proposed changes to the manuscript will make it an even more valuable contribution to the literature. I still recommend acceptance of this work.

---

> > > ### Author Response · Authors · 2025-08-05
> > >
> > > Thank you very much for your positive feedback. If you have any further questions or suggestions, please feel free to let us know.

---

### Official Review · Reviewer_dAst · 2025-07-01

**Clarity:** 3
**Significance:** 3
**Originality:** 3
**Rating:** 5
**Confidence:** 4

**Summary:**

This paper proposes a motion-induced object-centric pretraining method for vision foundation models. The proposed method uses an optical flow model to extract motion clues from raw videos, and derive object pseudo labels through clustering on optical flow map. Then a simple contrastive learning objective is proposed to learn the concept of object from the annotated images. Extensive experiments on depth estimation, 3D object detection and 3D occupancy prediction validate the effectiveness of the proposed method.

**Questions:**

1. Could the authors explain the similar performance with SimMIM in depth estimation and occupancy prediction?
2. Could the authors provide the results for SimMIM under smaller model sizes for more comprehensive comparison?
3. How does the method perform in other 2D downstream tasks, such as 2D semantic/instance segmentation?

**Ethical Concerns:**

["NO or VERY MINOR ethics concerns only"]

**Final Justification:**

Post rebuttal, the additional experiments have resolved my concerns about downstream tasks and performance. Thus I am leaning to accept the paper.

**Limitations:**

yes

**Quality:**

3

**Strengths And Weaknesses:**

Strengths:
1. Good motivation. The proposed method originates from the biological perspective and aligns with the development of human cognition.
2. Simple yet effective method. The proposed method consists of a simple contrastive loss, but is verified to surpass other advanced pretraining methods.
3. Extensive experiments. The paper conducts extensive experiments on three downstream tasks, on all of which the proposed method demonstrate good performance.

Weaknesses:
1. The performance of the proposed method is very close to SimMIM under large model parameters.
2. The types of downstream tasks is limited, all of which is closely related to 3D perception of the environment. According to the math derivation, the pseudo labels inherently correlate with the depth of the image, which might render 3D-related downstream tasks unfair for other pretraining methods.

---

> ### Author Rebuttal · Authors · 2025-07-31
>
> Dear Reviewer dAst,
> We greatly appreciate your valuable comments and questions. Our responses are as follows.
>
> ***Questions about SimMIM***
> For all methods compared in our paper, we utilize the official pretrained weights. Unfortunately, SimMIM does not provide models smaller than the Base version. Here we report results for the Base version of SimMIM on depth estimation.
>
> | Method | Backbone | SILog ↓ | Abs Rel ↓ | Sq Rel ↓ | RMSE ↓ | RMSE log ↓ | δ1 ↑ | δ2 ↑ | δ3 ↑ |
> | :---- | :---- | :---- | :---- | :---- | :---- | :---- | :---- | :---- | :---- |
> | SimMIM | Swin-B | 6.835 | 0.050 | 0.142 | 2.034 | 0.076 | 0.975 | 0.997 | 0.999 |
> | SimMIM  | Swin-L | 6.542 | 0.048 | 0.130 | 1.941 | 0.073 | 0.979 | 0.998 | 0.999 |
> | Ours | Swin-B | 6.598 | 0.048 | 0.131 | 1.939 | 0.073 | 0.981 | 0.997 | 0.999 |
> | Ours | Swin-L | 6.558 | 0.047 | 0.129 | 1.929 | 0.072 | 0.981 | 0.997 | 0.999 |
>
> SimMIM adopts a dense pretraining strategy, which shows advantages on tasks like depth estimation and occupancy prediction. However, on 3D object detection, our model significantly outperforms SimMIM.
>
> Our model leverages all available autonomous driving data. We believe that access to more diverse datasets could further improve its performance and generalization ability.
>
> ***How does the method perform in other 2D downstream tasks, such as 2D semantic/instance segmentation?***
> The following table presents the results of semantic segmentation using UperNet, 2D detection and instance segmentation using Mask R-CNN on the nuImages dataset.
>
> | Method | Backbone | mAP(detection bbox) | mAP(instance seg) | mIoU(semantic seg) |
> | :---- | :---- | :---- | :---- | :---- |
> | ImageNet | Swin-T | 53.1 | 44.3 | 85.74 |
> | Ours | Swin-T | 53.7 | 44.7 | 83.27 |
> | ImageNet | Swin-S | 53.9 | 44.8 | 87.16 |
> | Ours | Swin-S | 54.8 | 45.5 | 84.82 |
> | ImageNet | Swin-B | 54.4 | 45.1 | 87.51 |
> | Ours | Swin-B | 55.0 | 45.8 | 85.55 |
> | ImageNet | Swin-L | 54.7 | 45.3 | 88.49 |
> | Ours | Swin-L | 55.2 | 45.5 | 85.69 |
>
> As shown in the table, our method achieves comparable performance on 2D detection and instance segmentation tasks, but also shows noticeable shortcomings in semantic segmentation. Consequently, features learned by our model are instance-discriminative rather than category-discriminative. However, as demonstrated in paper's Table 3, our features are complementary to class-level features.

---

> ### Comment · Reviewer_dAst · 2025-08-05
>
> Thanks for the thoughtful response. The experiments have resolved my concerns about downstream tasks and performance. I will raise my score.

---

> > ### Author Response · Authors · 2025-08-08
> >
> > Thank you for your encouraging feedback and for taking the time to re-evaluate our work. If you have any further questions or suggestions, please feel free to let us know.

---

### Official Review · Reviewer_9fwV · 2025-07-02

**Clarity:** 3
**Significance:** 3
**Originality:** 3
**Rating:** 3
**Confidence:** 4

**Summary:**

The paper proposes to learn semantically separated features from motion signals. The method uses an optical flow model to predict the flow for a large set of driving scenes. This flow is then used in a clustering step to create pseudo-labels for all videos. A network is trained on those pseudo-labels and evaluated on various tasks, such as Depth Estimation, BEV Prediction, and Occlusion Prediction. The results show the method can be effective for these tasks.

**Questions:**

While I think the idea proposed by this paper is valuable, there are various aspects that need to be addressed.
- Extension beyond just driving scenes: How does the method behave when trained on other natural scenes?
- How does the approach behave under different parameter combinations & optical flow models (see above)?
- How does it compare to other video SSL approaches and other DINOv2 sizes?

If the authors can address my concerns, I'm willing to update my score.

**Ethical Concerns:**

["NO or VERY MINOR ethics concerns only"]

**Final Justification:**

While the authors did address some of my concerns convincingly, I remain in doubt about how general the approach can be and what the effect of the optical flow model is on the overall approach. I will keep my score of borderline reject.

Overall, the paper is on the right track, but needs some more improvements in the cited aspects.

**Limitations:**

Barely, but yes.

**Quality:**

3

**Strengths And Weaknesses:**

Strengths:
- The idea of exploiting coherent motion as as representation of objectness is very interesting and appealing
- The fact that the method is designed to work on a large unlabeled data pool of videos makes it scalable
- The presented quantitative results are somewhat consistent and improve upon the SOTA
- The qualitative results are nicely presented and show the effectiveness of the pseudo-labeling

Weaknesses:
- The choice of datasets seems to be restricted to driving videos. I wonder why this choice was made since the method could also work on other types of natural videos? This choice fails to fully exploit the potential of the proposed method
- The proposed method is rather simple and seems a bit handcrafted. However, it is somewhat effective, so this is not critical.
- Lack of ablations: In the main paper, the method’s components and parameter choices are barely ablated. I think this is crucial for the paper. There would be a lot of value in seeing how effective the pseudo-labeling is under different parameter choices like the manually chosen thresholds in L183, different optical flow models like RAFT [1] or SEA-RAFT [2], different choices for the margin parameter L209, and so on
- More model comparisons: It would be great to also see other video self-supervised models compared, for example [3]. Furthermore, DINOv2 was only compared with a ViT-S, while the proposed method uses up to a Swin-L. It would be nice to also see other sizes for DINOv2.

[1] Teed, Zachary, and Jia Deng. "Raft: Recurrent all-pairs field transforms for optical flow." Computer Vision–ECCV 2020: 16th European Conference, Glasgow, UK, August 23–28, 2020, Proceedings, Part II 16. Springer International Publishing, 2020.
[2] Wang, Yihan, Lahav Lipson, and Jia Deng. "Sea-raft: Simple, efficient, accurate raft for optical flow." European Conference on Computer Vision. Cham: Springer Nature Switzerland, 2024.
[3] Salehi, Mohammadreza, et al. "Time does tell: Self-supervised time-tuning of dense image representations." Proceedings of the IEEE/CVF International Conference on Computer Vision. 2023.

---

> ### Author Rebuttal · Authors · 2025-07-31
>
> Dear Reviewer 9fwV,
> We greatly appreciate your valuable comments and questions. Our responses are as follows.
>
> ***Extension beyond just driving scenes: How does the method behave when trained on other natural scenes?***
>
> We use driving data because current optical flow models perform better in such environments. In real-world datasets like Ego4D, where high-quality optical flow data is lacking, flow estimation remains unreliable. However, as discussed in paper's Sections 5.1 and 5.2, our method shows reasonable generalization to natural scenes.
>
> ***How does the approach behave under different parameter combinations & optical flow models?***
>
> The following table shows the depth estimation results of the Swin-T model after 200 epochs of pretraining with different values of $\\theta\_f$​.
>
> | $\\theta\_f$ | SILog ↓ | Abs Rel ↓ | Sq Rel ↓ | RMSE ↓ | RMSE log ↓ | δ1 ↑ | δ2 ↑ | δ3 ↑ |
> | :---- | :---- | :---- | :---- | :---- | :---- | :---- | :---- | :---- |
> | 0.5 | 7.069 | 0.051 | 0.146 | 2.050 | 0.078 | 0.974 | 0.997 | 0.999 |
> | 1.5 | 7.022 | 0.051 | 0.146 | 2.051 | 0.079 | 0.974 | 0.997 | 0.999 |
>
> Here, we present the results of some downstream tasks with the margin set to 0.5, 1.0, and 1.5, trained for 100 epochs on one-quarter of the dataset using Swin-T as backbone.
>
> |  | BEVFormer v2 |  | Depth Estimation |  |  |  |
> | :---- | :---- | :---- | :---- | :---- | :---- | :---- |
> | **Margin** | **mAP ↑** | **NDS ↑** | **SILog ↓** | **Abs Rel ↓** | **Sq Rel ↓** | **RMSE ↓** |
> | 0.5 | 42.41 | 51.87 | 7.329 | 0.54 | 0.157 | 2.079 |
> | 1.0 | 42.81 | 51.72 | 7.284 | 0.53 | 0.153 | 2.090 |
> | 1.5 | 42.54 | 52.23 | 7.244 | 0.53 | 0.154 | 2.086 |
>
> According to the ablation study, our model demonstrates robustness to hyperparameter variations when trained on large-scale datasets, except in extreme cases—such as setting the margin to 0, which indicates infinite clusters.
>
> ---
>
> We chose VideoFlow specifically because it performs joint multi-frame estimation, which provides better temporal consistency compared to two-frame-based models such as RAFT. This is particularly beneficial for long-term motion tracking in videos.
>
> However, our method is not limited to a specific optical flow model — it can benefit from improvements in flow estimation. For example, we are actively exploring more robust approaches like CoTracker \[4\], which offer enhanced pixel-level tracking across time.
>
> ***How does it compare to other video SSL approaches and other DINOv2 sizes?***
> Additional results on DINOv2 are provided in the supplementary materials.
>
> We tested the TimeT\[3\] model on depth estimation and BevFormer v2(704 × 256 resolution). The official TimeT GitHub repository only offers the ViT-S model. The following table shows the comparison results for the Small model.
>
> |  |  | BEVFormer v2 |  | Depth Estimation |  |  |  |
> | :---- | :---- | :---- | :---- | :---- | :---- | :---- | :---- |
> | **Method** | **Backbone** | **mAP ↑** | **NDS ↑** | **SILog ↓** | **Abs Rel ↓** | **Sq Rel ↓** | **RMSE ↓** |
> | DINOv2 | ViT-S | 46.24 | 34.88 | 7.119 |  0.052 | 0.158 | 2.153 |
> | TimeT | ViT-S | 41.02 | 28.63 | 8.551 | 0.062 | 0.219 | 2.484 |
> | Ours | Swin-S | 50.87 | 40.23 | 6.736 | 0.049 | 0.138 | 1.981 |
>
> As shown in the table, since the TimeT model was not designed or trained for large-scale pretraining, it exhibits a significant gap compared to existing methods on both tasks.  Nevertheless, these works remain valuable, and we will include a discussion of them in the paper.
>
> \[4\] Karaev, Nikita, et al. Cotracker: It is better to track together. ECCV 2024.

---

> > ### Comment · Reviewer_9fwV · 2025-08-05
> > **Rebuttal Response**
> >
> > I appreciate the authors' response to my concerns. With the new experiments, some of these concerns like the ablations are addressed. However, some others remain unaddressed. And I would like to highlight that the paper in general is interesting and well presented, but is needs more technical improvements.
> >
> > One critical ablation the paper is missing is the comparison of different optical flow estimators to integrate into the method. This would enable a better understanding of how the performance of the flow estimator impacts the proposed method, which, in my opinion, is a crucial aspect.
> >
> > Also, I found the authors' answer about optical flow quality being best in driving scenarios as a reason for why natural scenes were not evaluated unconvincing. A set of recent unsupervised (moving) object segmentation papers [1,2,3] have relied on off-the-shelf optical flow predicted natural scenes such as the DAVIS dataset. While the authors' tackle a different task in their work, and therefore the results are not 1:1 translatable, these works show that the quality of optical flow on natural scenes is sufficient to learn video semantics without labels.
> >
> > As it is right now, I'm not convinced this paper is ready to be accepted, but I'm happy hear the authors' reiterated thoughts on the optical flow on natural scenes discussion.
> >
> >
> > [1] Xie, Junyu, Weidi Xie, and Andrew Zisserman. "Segmenting moving objects via an object-centric layered representation." Advances in neural information processing systems 35 (2022): 28023-28036.
> > [2] Karazija, Laurynas, et al. "Learning segmentation from point trajectories." Advances in Neural Information Processing Systems 37 (2024): 112573-112597.
> > [3] Xie, Junyu, et al. "Moving object segmentation: All you need is sam (and flow)." Proceedings of the Asian conference on computer vision. 2024.

---

### Official Review · Reviewer_6akB · 2025-07-03

**Clarity:** 3
**Significance:** 3
**Originality:** 4
**Rating:** 4
**Confidence:** 4

**Summary:**

The authors propose a framework to learn object-centric visual features from motion inspired by findings from developmental neuro-science. An off-the-shelf flow model is used to detect motion in videos. The flow is then clustered to generate object pseudo-labels. The labeled clusters are then used to train an encoder using contrastive loss.  The learnt features are evaluated on the tasks of depth-estimation, 3D object detection and occupancy prediction. State-of-the-art performance is demonstrated with respect to several relevant baselines.

**Questions:**

1. The clustering is done based on flow information. What if the data contains video with low motion/ or videos of scenes with no specific objects, how do instances like these affect the clustering?
2. Is there a curation process that the data has to go through to be able to generate reasonable clusters.

**Ethical Concerns:**

["NO or VERY MINOR ethics concerns only"]

**Final Justification:**

The rebuttal satisfactorily answers most concerns raised and the additional results help strengthen the claims of the paper.

**Limitations:**

The limitations of the framework have been adequately discussed

**Paper Formatting Concerns:**

Paper adheres to NeurIPS guidelines.

**Quality:**

3

**Strengths And Weaknesses:**

## Strengths:
1. **Label free method**: The proposed approach is label-free and does not require additional annotations like camera parameters. This is particularly useful to extract structured information from a large collection of videos.
2. **Quantitative results**: The approach outperforms DINOv2 and other relevant baselines on 3D tasks.


## Weaknesses:
1. **Novelty**: The core idea of using motion to learn objectness has been explored before (points on a object have 'common fate'). The approach uses an off the shelf flow estimation model
2. **Dependence on optical flow estimator**: The base of the approach relies on clustering the motion representation obtained from an off-the-shelf optical flow estimator thus inherits all the limitation of the chosen base model. In particular, optical flow models fail to do well on flat textured/ untextured regions of a video. Since the clustering and training starts from this step, is there a tendency for the errors to cascade across the stages?
3. **Qualitative examples**: Providing more qualitative results would be beneficial to help appreciate the effectiveness of the approach.
4. **Writing**: There are number of typos and language errors throughout the manuscript. The paper would benefit from a thorough proof reading and streamlining of certain sections.

---

> ### Author Rebuttal · Authors · 2025-07-31
>
> Dear Reviewer 6akB,
> We greatly appreciate your valuable comments and questions. Our responses are as follows.
>
> ***Novelty***
> Our idea builds upon numerous prior works on object discovery, and we did not intend to claim this as a novel insight. Our main contribution lies in demonstrating the effectiveness and scalability of this idea on a large-scale dataset and achieving comparable performance to foundation models such as DINO on downstream tasks.
> We will revise some sentence in our paper, for example:
> 1. "Our key insight is that motion boundary serves as a strong signal for object-level grouping." will be revised to “we were inspired by the insight that motion boundary serves as a strong signal for object-level grouping, which can be used to derive pseudo instance supervision from raw videos.”
> 2. "we are the first to leverage motion as an unsupervised supervisory signal to guide the emergence of object-level semantics in visual representations." We will revise it to “We are first to demonstrate the effectiveness and scalability of using motion as an unsupervised supervisory signal on a large-scale dataset and modern model architectures”
>
> ***Is there a curation process that the data has to go through to be able to generate reasonable clusters?***
>
> Since the nuPlan dataset provides camera distortion parameters, we applied undistortion to all images when processing the nuPlan data. Apart from this, all procedures are exactly as described in the paper(Lines 173–191) and Section A of the supplementary material, with no additional processing involved. We retained all samples with at least two pseudo-label(i.e. at least one foreground cluster) since samples that cluster all pixels into a single group have a loss of 0 and are not useful for training.
>
> ***The clustering is done based on flow information. What if the data contains video with low motion/ or videos of scenes with no specific objects, how do instances like these affect the clustering?***
>
> The video contains low motion or no objects which failed to cluster instances will be filtered out during preprocessing, as described in Lines 189–191. Given the large scale of the dataset, the model still learns generalizable representations, even with these less informative frames removed.
>
> ***Qualitative examples***
> Additional Qualitative results are provided in the supplementary material. Unfortunately, due to system limitations, we are unable to include more during the rebuttal phase.
>
> ***Writing***
> We appreciate your suggestions. We will continue to improve our paper and address these writing errors.

---

> > ### Comment · Reviewer_6akB · 2025-08-04
> > **Response to rebuttal**
> >
> > The rebuttal answers most of the concerns raised. Further, The new comparisons and additional analysis w.r.t different flow models and different downstream tasks helps strengthen the claims of the paper. To that effect, I am updating my score to be slightly more positive. The authors are encouraged to revise the writing and tone done certain claims w.r.t to discovering objectness from motion.

---

> > > ### Author Response · Authors · 2025-08-05
> > >
> > > We are glad to hear that we have addressed your concerns. We will revise our paper accordingly based on the reviewers' comments. If you have any further questions or suggestions, please feel free to let us know.

---

### Official Review · Reviewer_h9dF · 2025-07-09

**Clarity:** 3
**Significance:** 3
**Originality:** 3
**Rating:** 4
**Confidence:** 4

**Summary:**

The authors introduce a self-supervised, instance-aware framework for learning dense visual representations. Their supervision signal is derived from pseudo-segmentation maps obtained by clustering optical flow. An encoder is trained using a triplet ranking loss to produce dense feature maps where pixel-wise similarities align with the inferred region maps. For evaluation, the authors present region-similarity visualizations and quantitative benchmarks on depth estimation and 3D object detection, demonstrating that their approach outperforms contrastive-based self-supervised learning methods.

**Questions:**

I have questions about the robustness of the proposed methods for different flow estimations and their performance on the standard benchmark. Please see the weakness for details.

**Ethical Concerns:**

["NO or VERY MINOR ethics concerns only"]

**Final Justification:**

After reading the author's and other reviewers' comments, I kept my rating to weak acceptance. Learning instance-wise representation without any human supervision is a very challenging task, and the authors propose a simple and effective framework. I wish there will be following-up work to test this approach at larger scale and provide more extensive benchmark.

**Limitations:**

See weakness.

**Paper Formatting Concerns:**

No formatting concerns for this paper.

**Quality:**

2

**Strengths And Weaknesses:**

Strength:
(1) Learning instance-level representations in a self-supervised manner remains a challenging task. The authors present a simple yet effective framework that outperforms state-of-the-art contrastive learning approaches on selected benchmarks.
(2) Visualizations of the learned pixel-wise affinity maps indicate that the model captures meaningful instance-level embeddings.
(3) Good paper presentation.

Weakness:
(1) The method relies on externally estimated optical flow as input, but the authors do not include ablation studies across different flow estimators to assess the robustness of their approach.
(2) While the method is evaluated on depth estimation and 3D object detection benchmarks, it lacks several key metrics: specifically, evaluations of the learned feature maps on semantic and instance segmentation tasks via fine-tuning, linear probing, or unsupervised clustering. Comparisons to baseline methods, such as DINO, on image classification tasks are also missing.
(3) The backbone used in the proposed method differs from those in baseline approaches (e.g., DINO), as it is designed to produce higher-resolution feature maps. This architectural difference raises questions about whether performance gains in high-resolution tasks like depth estimation are due to the proposed method or simply the model architecture. For instance, could a DINO model with a similar FPN-style decoder yield comparable results?
(4) The paper omits comparisons with relevant prior work that also targets instance-aware embeddings [1] or learns objectness from video signals [2].

[1] Zhang et al., Self-Supervised Visual Representation Learning from Hierarchical Grouping
[2] Liu et al., The Emergence of Objectness: Learning Zero-Shot Segmentation from Videos

---

> ### Author Rebuttal · Authors · 2025-07-31
>
> Dear Reviewer h9dF,
> We greatly appreciate your valuable comments and questions. Our responses are as follows.
>
> ***Flow models***
>
> We chose VideoFlow specifically because it performs joint multi-frame estimation, which provides better temporal consistency compared to two-frame-based models such as RAFT. This is particularly beneficial for long-term motion tracking in videos.
>
> However, our method is not limited to a specific optical flow model — it can benefit from improvements in flow estimation. For example, we are actively exploring more robust approaches like CoTracker \[3\], which offer enhanced pixel-level tracking across time.
>
> ***Evaluations on other tasks***
> The following table presents the results of semantic segmentation using UperNet, 2D detection and instance segmentation using Mask R-CNN on the nuImages dataset.
>
> | Method | Backbone | mAP(detection bbox) | mAP(instance seg) | mIoU(semantic seg) |
> | :---- | :---- | :---- | :---- | :---- |
> | ImageNet | Swin-T | 53.1 | 44.3 | 85.74 |
> | Ours | Swin-T | 53.7 | 44.7 | 83.27 |
> | ImageNet | Swin-S | 53.9 | 44.8 | 87.16 |
> | Ours | Swin-S | 54.8 | 45.5 | 84.82 |
> | ImageNet | Swin-B | 54.4 | 45.1 | 87.51 |
> | Ours | Swin-B | 55.0 | 45.8 | 85.55 |
> | ImageNet | Swin-L | 54.7 | 45.3 | 88.49 |
> | Ours | Swin-L | 55.2 | 45.5 | 85.69 |
>
> We conducted a linear probe evaluation on Cityscapes dataset with ResNet-50 as backbone.
>
> | Method | mIoU |
> | :---- | :---- |
> | ImageNet | 44.14 |
> | MoCo v3 | 46.25 |
> | Ours | 33.33 |
> | Ours \+ ImageNet | 48.40 |
> | Ours \+ MoCo v3 | 49.50 |
>
> As shown in the table, our method achieves comparable performance on 2D detection and instance segmentation tasks, but also shows noticeable shortcomings in semantic segmentation. This is because the features learned by our model are instance-discriminative rather than category-discriminative. However, as demonstrated in paper’s Table 3 and the table above, our features are complementary to class-level features.
>
> ***The architectural difference***
> As described in lines 233–234, although our method is pretrained with an FPN, during fine-tuning on downstream tasks, only the pretrained weights of the backbone are used. This is because our pretraining task involves pixel-level supervision, which benefits from multi-scale features.  We also conducted additional experiments at lower resolutions, as shown in Table 2 of the supplementary material. The results demonstrate that our method remains effective even at lower resolutions.
>
> ***Comparisons with relevant works***
> We found it quite difficult to make fair comparisons with other relevant works under our experimental setup. Many works do not open-source their code \[1\] or fail to provide downloadable pretrained weights \[2\]. Nevertheless, these works remain valuable. We will include them in the discussion section of the paper and conduct comparisons once the official code is available.
> We tested a video self-supervised learning model TimeT\[4\] on depth estimation and BevFormer v2(704 × 256 resolution). The official TimeT GitHub repository only offers the ViT-S model. The following table shows the comparison results for the Small model.
>
> |  |  | BEVFormer v2 |  | Depth Estimation |  |  |  |
> | :---- | :---- | :---- | :---- | :---- | :---- | :---- | :---- |
> | **Method** | **Backbone** | **mAP ↑** | **NDS ↑** | **SILog ↓** | **Abs Rel ↓** | **Sq Rel ↓** | **RMSE ↓** |
> | DINOv2 | ViT-S | 46.24 | 34.88 | 7.119 |  0.052 | 0.158 | 2.153 |
> | TimeT | ViT-S | 41.02 | 28.63 | 8.551 | 0.062 | 0.219 | 2.484 |
> | Ours | Swin-S | 50.87 | 40.23 | 6.736 | 0.049 | 0.138 | 1.981 |
>
> As shown in the table, since the TimeT model was not designed or trained for large-scale pretraining, it exhibits a significant gap compared to existing methods on both tasks.
>
> \[1\] Zhang et al., Self-Supervised Visual Representation Learning from Hierarchical Grouping\. NeurIPS 2020
>
> \[2\] Liu et al., The Emergence of Objectness: Learning Zero-Shot Segmentation from Videos\. NeurIPS 2021
>
> \[3\] Karaev, Nikita, et al. Cotracker: It is better to track together. ECCV 2024.
>
> \[4\] Salehi, Mohammadreza, et al. Time does tell: Self-supervised time-tuning of dense image representations. ICCV. 2023.

---

> > ### Comment · Reviewer_h9dF · 2025-08-04
> > **Thank you for your response.**
> >
> > Thank you for providing extra experiments. Even though I was inclined to accept the paper in my initial review but I would like to see the linear probing performance on a standard benchmark, e.g., ImageNet-1k and compare to other feature learning model.

---

> > > ### Author Response · Authors · 2025-08-08
> > >
> > > Thank you for your valuable comment.
> > >
> > > Here we report linear-probe results on the ImageNet dataset. As expected, the performance is notably lower than DINO. We attribute this to two main factors:
> > >
> > > 1. Our approach is designed to learn **instance-level** rather than **category-level** semantics.
> > >
> > > 2. The current model is trained solely on outdoor driving videos, making ImageNet—a dataset with vastly different visual statistics—an out-of-domain benchmark.
> > >
> > > Interestingly, the smallest backbone (Swin-T) achieves the highest accuracy among our variants, further suggesting that the current ImageNet setting lies far outside our training distribution.
> > > As noted in our paper, we plan to expand the diversity of our training data to build a more generalizable instance-centric representation model in future work.
> > >
> > > | Backbone | Pretrain | accuracy/top1 | accuracy/top5 |
> > > | :--- | :--- | :--- | :--- |
> > > | Swin-T | Ours | 31.98 | 54.78 |
> > > | Swin-S | Ours | 20.46 | 39.93 |
> > > | Swin-B | Ours | 20.17 | 38.95 |
> > > | Swin-L | Ours | 18.52 | 37.19 |
> > > | Swin-B | SimMIM | 27.85 | 49.49 |
> > > | Swin-L | SimMIM | 34.84 | 56.77 |
> > > | ViT-B | DinoV2 | 83.34 | 96.24 |
> > > | ViT-B | MAE | 66.99 | 86.77 |
> > > | ViT-B | MoCoV3 | 76.28 | 92.75 |

---

### Comment · Area_Chair_n7cW · 2025-08-02

Dear 7845 Reviewers: The authors have provided detailed rebuttals to your reviews. I'd urge you to read their rebuttals (as well as other reviews) early to allow further interactions that help clarify any lingering confusion or misunderstanding. Thank you! AC

---

### Decision · Program_Chairs · 2025-09-17

**Decision:**

Accept (poster)

**Comment:**

The paper introduces a self-supervised, object-centric representation learning method that uses optical flow clustering to generate pseudo-instance masks for training a dense encoder via contrastive loss. Inspired by developmental psychology, the method learns pixel-wise embeddings aligned with motion-derived regions from unlabeled videos. It is evaluated on downstream 3D tasks such as monocular depth estimation, 3D object detection, and occupancy prediction, demonstrating strong performance and outperforming both supervised and self-supervised baselines.

Reviewers praised the method’s simplicity, effectiveness, and scalability, noting that it leverages motion cues in a biologically inspired way. The label-free approach performs well across tasks and surpasses prior methods like SimMIM and DINOv2. Visualizations of learned pixel affinities were compelling, and one reviewer highlighted its generalization beyond driving scenes. The work was seen as timely, with potential for broader impact in unsupervised video representation learning.

Key concerns include the method’s reliance on external optical flow, with no ablations on flow models or sensitivity to thresholds and hyperparameters. Evaluation focused mostly on 3D driving datasets, leaving open questions about generalizability to 2D or non-motion tasks. Some novelty claims were seen as exaggerated, and differences in model backbones raised fairness concerns in comparisons. Reviewers also noted missing evaluations (e.g., segmentation, linear probes), limited related work citations, and called for clearer writing and terminology.

The authors’ rebuttal addressed many core concerns, including ablations and architectural comparisons, and the additional experiments on 3D downstream tasks helped solidify the method’s empirical claims. Reviewers noted the strong performance across multiple benchmarks and appreciated the paper’s clarity and potential for follow-up work.

However, one reviewer maintained a borderline reject, expressing concern over the method’s limited generalization beyond driving scenes, which restricts its broader applicability. While the authors defended their domain choice by citing optical flow quality, the reviewer found this reasoning unconvincing given recent work using natural scene videos (non-driving datasets like DAVIS) with off-the-shelf flow. Additionally, the lack of ablations on different optical flow estimators was seen as a missed opportunity to understand the method’s robustness.  These concerns suggest that while the framework is promising, its broader applicability beyond driving scenarios is still uncertain.

In terms of novelty, the authors clarified during the rebuttal that their intent was not to claim motion boundaries as a new insight, but rather to demonstrate the scalability and modern implementation of this classical idea. They acknowledged prior work on motion-based object discovery and committed to revising phrasing in the paper to more accurately reflect this, e.g., changing “our key insight” to “we were inspired by the insight...” and reframing their novelty as demonstrating effectiveness on large-scale datasets with modern architectures. With these adjustments, and given the paper’s solid technical execution and relevance, the consensus (out of 2x accepts, 2x borderline accepts, 1x borderline reject) is to accept.